# Disentangling the contribution of hospitals and municipalities for understanding patient level differences in one-year mortality risk after hip-fracture: A cross-classified multilevel analysis in Sweden

Pia Kjær Kristensen [1,2☯*], Raquel Perez-Vicente[3], George Leckie[4], Søren Paaske Johnsen[5], Juan Merlo[3☯]

**1** Department of Clinical Epidemiology, Aarhus University Hospital, Aarhus, Denmark, **2** Department of Orthopedic Surgery, Regional Hospital Horsens, Horsens, Denmark, **3** Research Unit of Social Epidemiology, Clinical Research Centre, Faculty of Medicine, Lund University, Malmö, Sweden, **4** Centre for Multilevel Modelling, School of Education, University of Bristol, United Kingdom, **5** Danish Center for Clinical Health Services Research, Department of Clinical Medicine, Aalborg University, Aalborg, Denmark

☯ These authors contributed equally to this work.
\* pkkr@clin.au.dk

**Data Availability Statement:** Data cannot be shared publicly because of ethical and data safety

## Abstract

### Background

One-year mortality after hip-fracture is a widely used outcome measure when comparing hospital care performance. However, traditional analyses do not explicitly consider the referral of patients to municipality care after just a few days of hospitalization. Furthermore, traditional analyses investigates hospital (or municipality) variation in patient outcomes in isolation rather than as a component of the underlying patient variation. We therefore aimed to extend the traditional approach to simultaneously estimate both case-mix adjusted hospital and municipality comparisons in order to disentangle the amount of the total patient variation in clinical outcomes that was attributable to the hospital and municipality level, respectively.

### Methods

We determined 1-year mortality risk in patients aged 65 or above with hip fractures registered in Sweden between 2011 and 2014. We performed cross-classified multilevel analysis with 54,999 patients nested within 54 hospitals and 290 municipalities. We adjusted for individual demographic, socioeconomic and clinical characteristics. To quantify the size of the hospital and municipality variation we calculated the variance partition coefficient (VPC) and the area under the receiver operator characteristic curve (AUC).

### Results

The overall 1-year mortality rate was 25.1%. The case-mix adjusted rates varied from 21.7% to 26.5% for the 54 hospitals, and from 18.9% to 29.5% for the 290 municipalities.

reasons. Data are available from to the Swedish National Board of Health and Welfare and Statistics Sweden after approval of the research project by an Ethical Committee and by the data safety committees of the Swedish Authorities for researchers who meet the criteria for access to confidential data. The database was constructed after approval from the Ethical Committee in Sweden (https://etikprovningsmyndigheten.se/) and from the data safety committees of the Swedish National Board of Health and Welfare (https://www.socialstyrelsen.se/) and Statistics Sweden (https://www.scb.se/en/).

**Funding:** Juan Merlo - Swedish Research Council (i.e., Vetenskapsrådet) project-id: 2017-01321. https://www.swecris.se/betasearch/details/project/201701321VR Pia Kjær Kristensen- The Health Research Fund of Central Denmark Region A869. https://www.rm.dk/sundhed/faginfo/forskning/region-midtjyllands-sundhedsvidenskabelige-forskningsfond/ The funders had no role in study design, data collection and analysis, decision to publish, or preparation of the manuscript.

**Competing interests:** The authors have declared that no competing interests exist.

The VPC was just 0.2% for the hospital and just 0.1% for the municipality level. Patient sociodemographic and clinical characteristics were strong predictors of 1-year mortality (AUC = 0.716), but adding the hospital and municipality levels in the cross-classified model had a minor influence (AUC = 0.718).

## Conclusions

Overall in Sweden, one-year mortality after hip-fracture is rather high. However, only a minor part of the patient variation is explained by the hospital and municipality levels. Therefore, a possible intervention should be nation-wide rather than directed to specific hospitals or municipalities.

## Introduction

Increasing demands for healthcare fueled by aging populations, increasing prevalence of multimorbid patients and technological advances combined with restricted financial resources constitute a major challenge for health care systems globally [1]. To address this challenging task many countries have started to streamline their healthcare systems by improving care and reducing length of stay in hospitals, which includes earlier discharge to care in community settings or at home with professional support (i.e., homecare) [2, 3]. Hip fracture care is an example of this development where surgery is performed at the hospital while postoperative rehabilitation and further care of potential sequelae are done in the primary health care sector. Therefore, evaluating the quality of care after hip fracture needs to simultaneously consider both the hospital and the municipality settings. When evaluating quality of care after hip fracture, one-year mortality is frequently used as an outcome quality indicator in many healthcare systems [4]. From this perspective, identifying unwarranted variability between not only hospitals but also municipalities has considerable health policy relevance [5–7].

It is well-established that multilevel modelling is a suitable methodology when comparing different types of cluster (e.g., hospital, municipalities) for both statistical and conceptual reasons [8–18]. However, existing multilevel model studies evaluating healthcare performance after hip fracture have primarily focused on hospital level variation [12, 19–22]. Less work has been carried out analysing the potential variability among municipalities [23]. Cross-classified multilevel analysis allows us to decompose the total individual level variation into its hospital and its municipality level components [24, 25]. To our knowledge no previous studies have applied measures of components of variance and of discriminatory accuracy to examine individual variation in one-year mortality with respect to both hospitals and municipalities. Therefore, we use cross-classified multilevel analysis to evaluate one-year mortality in all patients suffering from hip fracture in Sweden.

## Population and methods

### Databases and study population

This historical follow-up study was based on prospectively collected data available from medical registries in Sweden (about 9.3 million inhabitants by 2010) with free access to medical care. At birth or upon immigration, all citizens in Sweden are assigned a unique registration number through which all contact with the healthcare system is recorded. This allows unambiguous record linkage between registries.

We obtained information on diagnoses, medication use and mortality from the Swedish Patient Register, the Swedish Prescribed Drug Register [26], and from the Cause of Death Register [27] administered by the National Board of Health and Welfare. We obtained demographic and socioeconomic information from the Population Register [28] and the Longitudinal Integration Database for Health Insurance and Labour Market Studies (LISA) register (http://www.scb.se/lisa), which are administrated by Statistics Sweden. To ensure the anonymity of the subjects, the Swedish authorities transformed the personal identification numbers of the individuals into arbitrary numbers before delivering the research databases to us, and we linked the databases using the anonymized identification number.

We defined hip fracture as the presence of any discharge diagnosis with a fracture of the femur coded according the International Classification of Diseases, 10[th] revision (ICD codes) as fracture of the femoral neck (S72.0), pertrochanteric (S72.1) or subtrochanteric (S72.2). From the Swedish Patient Registry, we identified all 56,161 patients being 65 years or older and residing in Sweden by 31th December 2010 who were discharged from the Swedish hospitals between 2011 and 2014 with a diagnosis of hip (i.e., femoral) fracture and surgery code. Next, we excluded 1,162 patients because they were residing less than five years in Sweden (n = 173), they have missing information on education (n = 976) or erroneous information on death date (n = 2). We also excluded 11 patients treated at facilities with less than 10 hip fracture patients during the study period (Fig 1). Patients which previous hip fracture compose of 11.95% (6,575/48,424) of our population.

## Ethics and data accessibility

The Regional Ethics Review Board in southern Sweden (Dnr: 2014/856) as well as the data safety committees from the National Board of Health and Welfare and from Statistics Sweden approved the construction of the database used in this study.

The database we analysed is not publicly available for ethical and data safety reasons. However, the same dataset can be constructed by request to the Swedish National Board of Health and Welfare and Statistics Sweden after approval of the research project by an Ethical Committee and by the data safety committees of the Swedish Authorities. The study also needs to be performed in collaboration with Swedish researchers. [29]

## Assessment of variables

**Mortality.**   For each patient we ascertained all-cause mortality within one year from the admission date to hospital.

**Sociodemographic and clinical characteristics of the patients.**   Part of the initial differences between hospitals or municipalities may relate to differences in case mix. To make the observational measurement of hospital and municipality effects as valid as possible we therefore adjusted for potential differences in patient sociodemographic and clinical characteristics as well as use of medication (Table 1).

**Sociodemographic characteristics.**   We use the age of the individuals in years as a continuous variable and included it in the analysis as a quadratic function. We categorized the patients as immigrants if they were born in another country and as natives if they were born in Sweden.

We calculated household individualized disposable income by dividing the total disposable income of the family of the patient by the number of family members, considering the different consumption weights of adults and children, according to Statistics Sweden (http://www.scb.se/lisa). We did so for the complete Swedish population in three occasions 2010, 2005 and 2000, and summarized the three occasion by computing the cumulative income. Finally, using the tertile values of the cumulative income distribution we divided the study population into

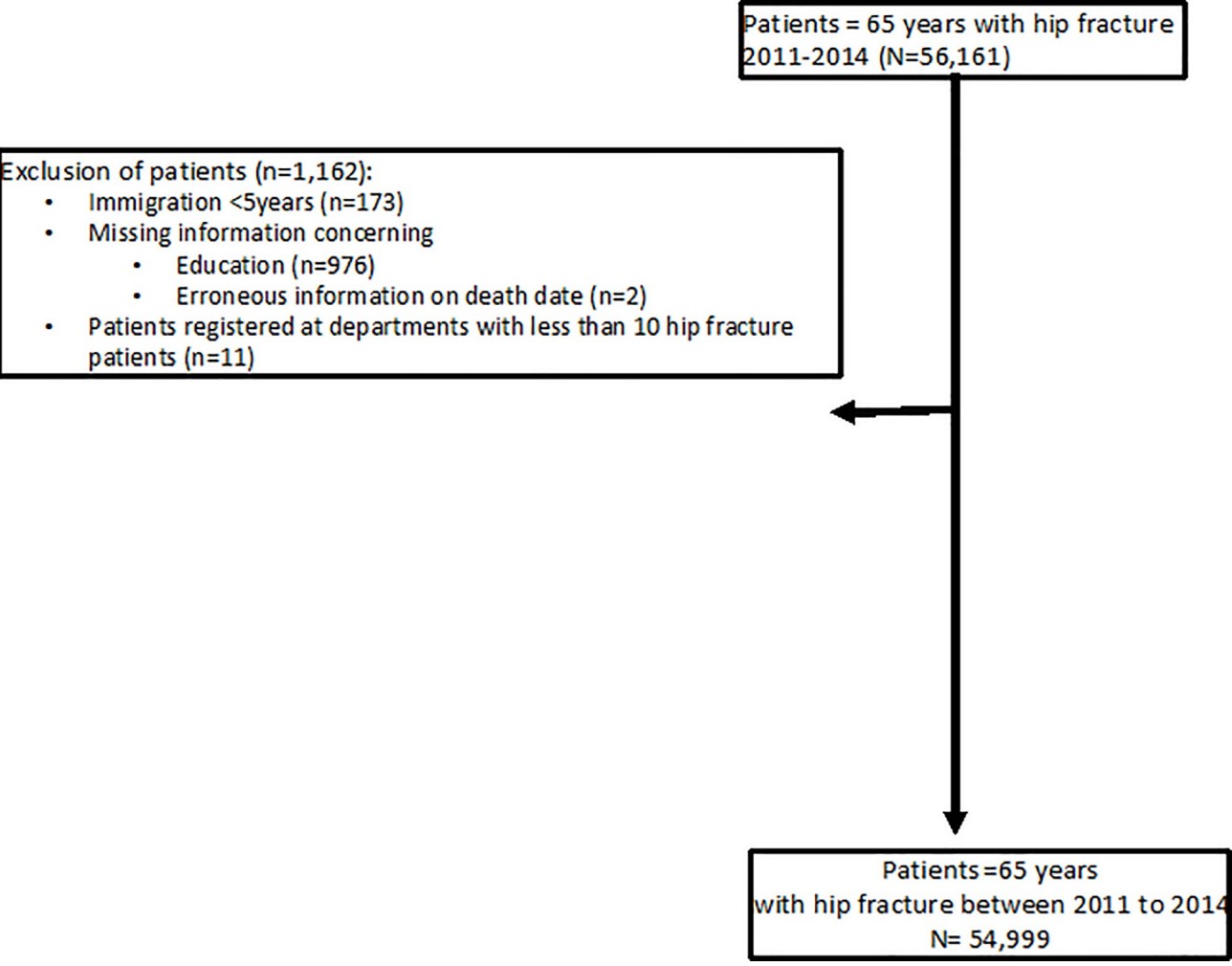

**Fig 1. Flowchart patient inclusion.**

high, medium and low income. In this form, we categorized each patient according his/her income category in the total population rather than the population of patients.

We classified education achievement as compulsory schooling of nine years or less, secondary education three years or less, and higher education. We defined low education achievement as compulsory school or less and use the value of the member of the family with the highest educational achievement for all member in the same family.

For defining those patients living alone, we first identified all patients who were cohabiting. That is, married couple, in a registered partnership or in an unregistered partnership with a common child as well as those living in a household with at least another adult. We group all other patients into the living alone category.

We combined sociodemographic characteristics into a single sociodemographic risk score for all-cause mortality. We did this via a logistic regression analysis modelling all-cause mortality as a function of age, gender, income, cohabiting, born in Sweden and education (see S1 Table). The individual predicted probabilities were then categorized into four groups by quartiles as low, medium, high, and very high. The low-risk score group was then used as the reference in the comparisons in the subsequent multilevel analyses.

**Table 1. Characteristic of the hip fracture population.** Values are percentages (number of patients) if not otherwise indicated.

| | |
|---|---|
| **Overall 1-year mortality** | 25.08 |
| **Number of patients in the population** | 54,999 |
| **Number of hospitals** | 54 |
| **Number of municipalities** | 290 |
| **Median number of patients at the hospital (min–max)** | 1015.50 (158–3,724) |
| **Median number of patients at the municipality (min–max)** | 189.65 (15–4,687) |
| **Age group (years)** | |
| • 65–74 | 18.38 (10,108) |
| • 75–84 | 42.29 (23,258) |
| • > 85 | 39.33 (21,633) |
| **Gender** | |
| • Men | 30.87 (16,976) |
| • Women | 69.13 (38,023) |
| Biomedical risk score for all-cause mortality | |
| • Low | 38.42 (21,132) |
| • Medium | 11.94 (6,566) |
| • High | 24.81 (13,645) |
| • Very high | 24.83 (13,656) |
| **Education** | |
| • Low education | 85.31 (46,922) |
| • High education | 14.69 (8,077) |
| **Income** | |
| • Low | 35.87 (19,728) |
| • Medium | 41.33 (22,731) |
| • High | 22.80 (12,540) |
| **Migration status** | |
| • Immigrant | 7.94 (4,365) |
| • Native (reference) | 92.06 (50,634) |
| **Cohabiting status** | |
| • Living alone (reference) | 64.76 (35,620) |
| • Living together | 35.24 (19,379) |
| **Medication** | |
| • Bisphosphonates | 0.63 (346) |
| • Analgesics | 27.07 (14,886) |
| • Psycholeptics | 59.90 (32,942) |
| • Psychoanaleptics | 43.53 (23,939) |

**Clinical characteristics.**   For the purpose of patient-mix adjustment and using previous knowledge on risk factors for mortality in patients with hip fractures [30, 31] as well as considering the variables included in the Charlson comorbidity index (CCI), we selected a number of diseases (ICD-10 codes) identified during the five years before hospital admission (see S2 Table). We then applied logistic regression to obtain a clinical risk score (i.e., individual predicted probability) of one-year mortality based on the following variables: Chronic kidney disease (N18), Acute myocardial infarction (I21), Heart failure (I50), Other peripheral vascular diseases (I73), Cerebrovascular diseases (I60-I69), Atherosclerosis- Aortic aneurysm and dissection (I70-I71), Dementia (F01-F03), Chronic lower respiratory diseases (J40-J47), Other disorders of the skin and subcutaneous tissue (L80-L99), Peptic ulcer (K27), Diseases of liver

(K70-K77), Diabetes mellitus (E08-E13), Hemiplegia (G81), Neoplasms (C00-D49), Human Immunodeficiency Virus (HIV) (B20), osteoporosis (M80-M81), previous Hip fracture (S70-S72). Finally, we created four categories of clinical risk scores using the quartile values of the risk score distribution and considering the group with the lowest risk score as reference in the comparisons.

**Use of medication.** Using the Anatomical Therapeutic Chemical (ATC) classification from the Swedish Prescribed Drug Register we also obtained information on previous use of Analgesics (ATC cod: N02), Psycholeptics (N05), Psychoanaleptics (N06) and Bisphosphonates (M05BA) as these medications may influence the vital prognosis of the patients [32, 33].

## Statistical analysis

We applied single-level (conventaional) and multilevel logistic regression analyses of discriminatory accuracy as decribed elsewhere [18]. We developed three consecutive logistic regression analyses to model one-year mortality.

The first model (model 1) is a single-level logistic regression including the socioeconomic risk score in four groups. This model aimed to evaluate the influence of patients' demographical characteristic on one-year mortality.

The second (model 2) is also a single-level logistic regression that expands model 1 to include the risk score for clinical factors as well as use of medication.

In the final, third model (model 3), we expanded model 2 by adding two random effects, one for the hospital level and the other for municipality level. In this way the model was converted into a two-way cross-classified multilevel model with the patients nested within the 54 hospitals and the 290 municipalities. As described elsewhere [18], we calculated the hospital and municipality general contextual effects as expressed by the variance partition coefficient (VPC) which informs on the share of the total individual variance in the propensity of one-year mortality that is at the hospital ($VPC_H$) and at the municipality level ($VPC_M$).

For all models, we calculated the Area Under the receiving operator characteristics Curve (AUC) or C-statistics [34] as a measure of discriminatory accuracy. Hosmer and Lemeshow [35] suggest that an AUC of 0.70 to 0.80 could be considered as 'acceptable', 0.80 to 0.90 as 'excellent' and 0.90 or above as 'outstanding', while an area under the ROC curve of 0.50 suggests no discrimination between the outcome groups (i.e., similar as tossing a coin to decide group membership). In model 3 we separately calculated the AUC when adding the random effect for the hospital ($AUC_H$) and the random effect for the municipality ($AUC_H$). Then, we calculated the increment in the AUC [18] when going from model 1 to model 2 ($AUC_{\Delta 2-1}$) and from model 2 to model 3 ($AUC_{\Delta 3-2}$). The increment in AUC measures the improvement in the ability of the model to correctly classify individuals with or without the outcome (i.e., one-year mortality) when considering the hospital or the municipality of the patients. The AUC provides analogous information as the VPC [24] as both measures inform on the *general contextual effect* of the hospital and of the municipality levels in relation to the patients' one-year mortality. As it has been explained elsewhere [12, 13, 18, 36–41] the general contextual effect, expresses the relevance of the hospital/municipality context for understanding patients' differences in one-year mortality.

In model 3 we use the predicted hospital and municipality random effects (shrunken residuals) to obtain case-mix adjusted and reliability-weighted average mortality rates and their 95% credible intervals and created league tables illustrating the ranking of the hospitals and of the municipalities.

In the supplementary material S1 File we provided an extended explanation of the methodology.

## Results

### Characteristics of the hip fracture population

The Swedish cohort of hip fracture patients consists of 54,999 patients. The unadjusted population one-year mortality was 25.1%. The hip fracture patients were mainly women, and most of the patients were living alone. Table 1 describes the additional characteristics of the population.

### Patient effects

High sociodemographic scores and high clinical scores were strong predictors for one-year mortality (Table 2). The AUC1 in model 1, which informs on the discriminatory accuracy of the sociodemographic information, had a value of 0.667 (95% CI: 0.662–0.672). Including the clinical score of the patients and use of medicines increased the AUC to 0.716 (95% CI: 0.711–0.720) (model 2). The use of analgesics, psycholeptics or psychoanaleptics was associated with

**Table 2. Analysis of 1-year mortality after hip fracture in the Swedish hospitals.**

| | Simple logistic regression analysis | | | | Cross classified multilevel logistic regression analysis | |
|---|---|---|---|---|---|---|
| | Model 1 | | Model 2 | | Model 3 | |
| **Specific individual average effects** | | | | | | |
| Sociodemographic RS | | | | | | |
| •Low | 1.00 | | 1.00 | | 1.00 | |
| •Medium | 1.88 | (1.75–2.01) | 1.74 | (1.62–1.86) | 1.71 | (1.59–1.82) |
| •High | 2.99 | (2.80–3.19) | 2.72 | (2.55–2.91) | 2.68 | (2.52–2.84) |
| •Very high | 5.68 | (5.33–6.04) | 5.34 | (5.01–5.69) | 5.29 | (4.98–5.57) |
| Clinical RS | | | | | | |
| •Low | | | 1.00 | | 1.00 | |
| •Medium | | | 1.18 | (1.10–1.27) | 1.17 | (1.08–1.25) |
| •High | | | 1.54 | (1.45–1.62) | 1.53 | (1.45–1.61) |
| •Very high | | | 2.67 | (2.53–2.81) | 2.66 | (2.52–2.80) |
| **Medication** | | | | | | |
| Bisphosphonates | | | 0.91 | (0.69–1.19) | 0.91 | (0.67–1.17) |
| Analgesics | | | 1.14 | (1.08–1.20) | 1.13 | (1.07–1.19) |
| Psycholeptics | | | 1.26 | (1.20–1.32) | 1.26 | (1.20–1.32) |
| Psychoanaleptics | | | 1.32 | (1.26–1.38) | 1.32 | (1.27–1.37) |
| **General contextual effects** | | | | | | |
| Hospital variance | | | | | 0.007 | (0.002–0.013) |
| Municipality variance | | | | | 0.002 | (0.001–0.005) |
| $VPC_H$ hospital | | | | | 0.2 | |
| $VPC_M$ municipality | | | | | 0.1 | |
| AUC | 0.667 | (0.662–0.672) | 0.716 | (0.711–0.720) | 0.718 | (0.713–0.722) |
| $AUC_{\Delta2-1}$ (increment model 2- model 1) | Reference | | 0.049 | | | |
| $AUC_{\Delta3-2}$ (increment model 3- model 2) | | | Reference | | 0.002 | |

RS = Risk score for all-cause mortality, VPC = Variance Partition Coefficient, AUC = Area under the receiver operating characteristic curve

1) Model 1: Simple logistic regression model including the socioeconomic risk score for all-cause mortality

2) Model 2: Simple logistic regression model including the socioeconomic and biomedical risk scores for all-cause mortality

3) Model 3: Cross-classified multilevel logistic regression model including the socioeconomic and biomedical risk scores for all-cause mortality and the hospitals and municipalities as random effects.

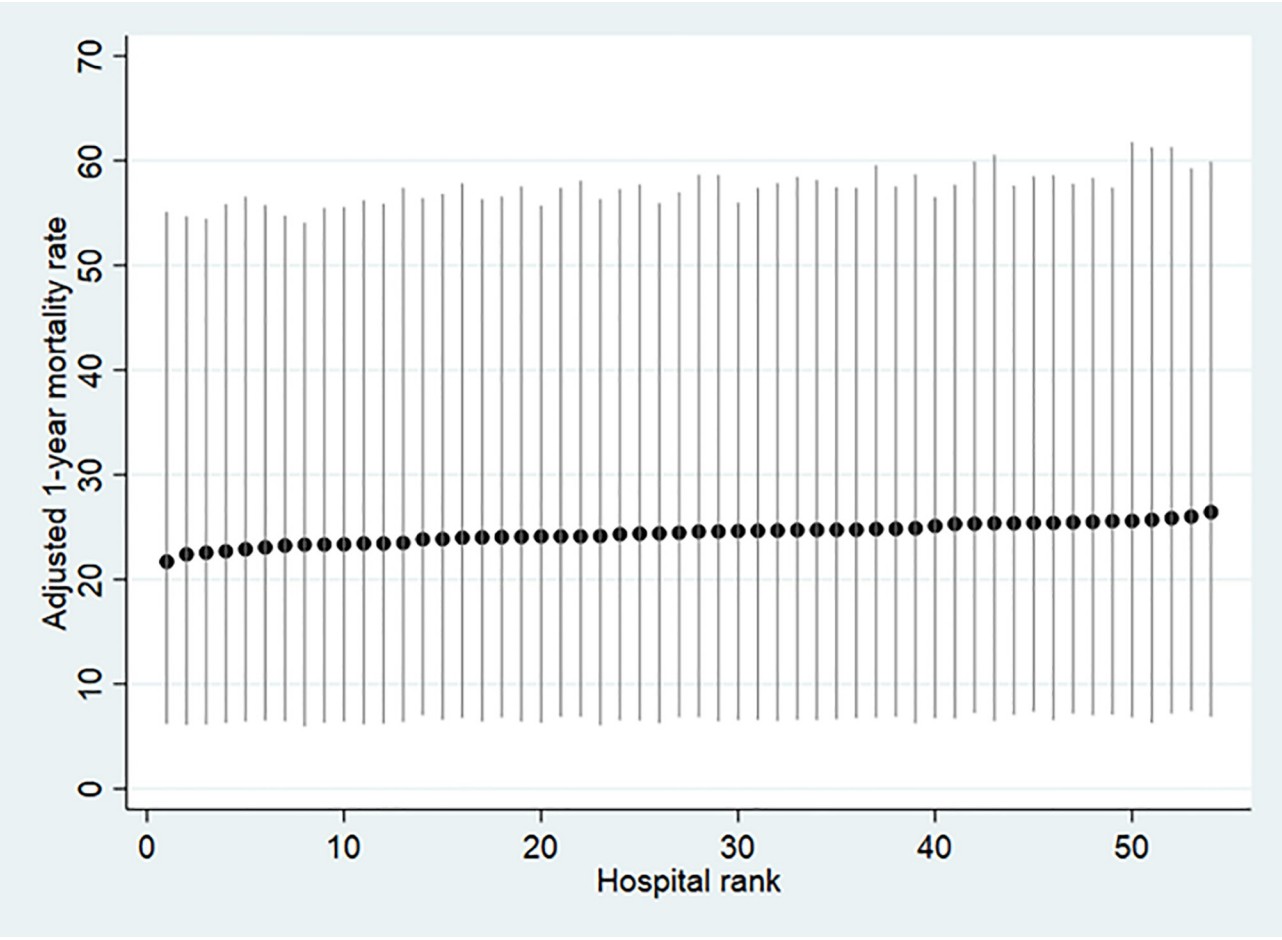

**Fig 2. League table ranking the 54 hospitals according to their adjusted absolute risk of 1-year mortality with 95% confidence intervals obtained from the cross-classified multilevel model.**

an increased risk of one-year mortality (Table 2). However, use of bisphosphonates was not associated with one-year mortality.

## Hospital effects

In the period from 2011 to 2014, 54 hospitals treating patients with hip fracture were included in our study. These hospitals treated between 158 and 3,724 patients. The hospital unadjusted one-year mortality expressed as percentage varied between 19.6% and 29.8%. The league table obtained from the multilevel analysis shows that after adjustment for patient case-mix and municipality effects the hospital mortality rates extended from 21.7% to 26.5% (Fig 2). The general contextual effect of the hospital on the patients' risk for one-year mortality was, however, low as only 0.2% of the adjusted individual variation in one-year mortality laid at the hospital level. Also, the increase in the discriminative accuracy when adding the hospital level to the model including only patient level variables was only 0.002 units (model 2).

## Municipality effects

After admission the hip fracture patients were discharged to 290 different municipalities in Sweden. The number of hip fracture patients in each municipality varied from 15 to 4,687 hip

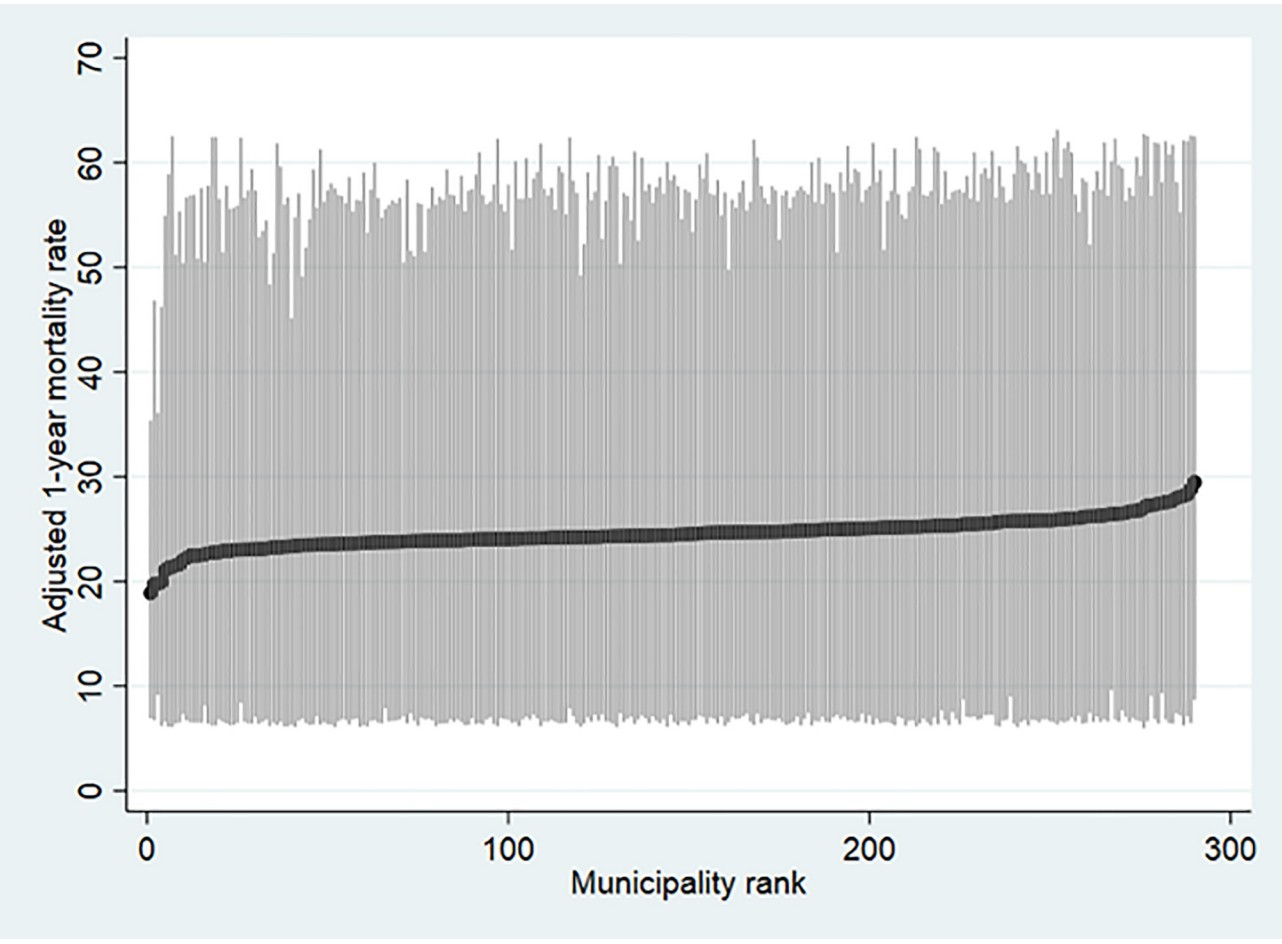

**Fig 3. League table ranking the 290 municipalities according to their adjusted absolute risk of 1-year mortality with 95% confidence intervals obtained from the cross-classified multilevel model.**

fracture patients in the three years. The unadjusted mortality rate expressed as percentage varies between 7.1% and 37.9%. After adjustment for the difference in case-mix, and the hospital effects in the multilevel analysis the municipality differences in one-year mortality extended between 18.9% and 29.5% (Fig 3). The general contextual effect of the municipalities on the patients' risk for one-year mortality was also low as only 0.1% of the total individual variance in the propensity of dying was at the municipality level. Also, the increase in the discriminative accuracy when adding the municipality level to the model including only patient level variables was only 0.002 units (model 2).

## Discussion

In this nationwide population-based study of Swedish hip fracture patients, the crude one-year mortality rate was 25.1% during the period 2011–2014. Patient sociodemographic and clinical characteristics were predictors of one-year mortality (AUC = 0.718), whereas the cross-classified multilevel analysis revealed that the AUC of the regressions model including hospitals and municipality as random effects was only marginally higher compared to the single-level model. Accordingly, hospital and municipality level variation corresponded to less than 1% of the overall individual variation in the underlying propensity of death within one year.

For evaluating the performance of the hospital and municipality health care performance we considered two measures, the overall unadjusted one-year mortality rate after hip fracture and the size of the general contextual effects. In our study the overall unadjusted one-year mortality rate of 25.08% is comparable to a previous Danish register study, which in the period 2011 to 2014 reported a one-year mortality of 26% [42]. However, our mortality rate was higher than that found in a previous study on seven European countries, which found a mortality rate of 22.3% in Sweden in 2007 [43]. In this European study the lowest mortality, 19%, was observed in Italy. Those differences might be explained by differences in study design and changing comorbidity patterns across time as well as country differences in healthcare provision [42, 44].

After adjustment for patient case-mix, the size of the general context effect is assumed to inform on the influence of the hospital and municipality health care levels on the vital prognosis of the patients [24]. The VPCs we found in this study were small, being 0.1% for municipalities and 0.2% for hospitals. Meaning that once we have adjusted for patient characteristics, patients treated at the same hospital or at the same municipality have very little in common according to their propensity of death within one year. That is, there were very small differences between hospitals and between municipalities (se elsewhere for an extended explanation on this concept [24])

To our knowledge no previous studies have applied cross-classified multilevel analyses to hip fracture data and so a direct comparison is therefore not possible. However, our results are in line with a previous Danish multilevel study examining hospital variation in 30-day mortality among hip fracture patient (but ignoring municipality variation), which found a VPC for hospitals of 0.9%.

The AUC of the cross-classified multilevel analysis was slightly higher compared to the single-level model, which still indicated that the cross-classified multilevel analysis was a better model than the single-level model even though the VPC was low for both the hospital and the municipality levels. However, the AUCs of the models were overall moderate, ranging from 0.667 to 0.718.

In our study bisphosphonates were not associated with reduced mortality. In randomized controlled trials, use of bisphosphonates have been shown to be effective for preventing both osteoporotic fractures [45–47] and mortality after hip fracture [48]. The protective effect of bisphosphonates on mortality after hip fracture has also been observed in numerous observational studies. However, recent research has suggested that the reduction in mortality occurs within the first weeks after treatment and, therefore, could express confounding [33].

Our study was strengthened by the cross-classified multilevel study design, as well as by the nationwide population-based coverage with complete follow-up on one-year mortality due to linkage to Swedish registries. However, our study only analysed the Swedish hospitals and municipalities and, therefore, cannot be generalized to other country contexts.

The treatment, care and rehabilitation of hip fracture patients at both hospitals and municipalities are fundamentally important. We found that hospital and mortality differences in average mortality risks were only a minor share of the total individual variance in the propensity to die or that, in other words, adding the hospital and municipality level does not increase the discriminatory accuracy obtained by patient level information only. This means that the hospital and municipality performances are homogenous overall in the country. Therefore, our results indicate that we will not lower mortality after hip fracture in Sweden by focusing on specific hospitals or specific municipalities with high average mortality. Rather, special efforts to reduce mortality after hip fracture should be focused on vulnerable patient groups of hip fracture patients wherever they are. From this reason, however, hospitals and municipalities with a higher number of vulnerable patients than other will need more intense interventions. Future

multilevel studies among hip fracture patients need to investigate differences in provided care and intermediate outcomes such as complications [49].

In conclusion, using cross-classified multilevel regression analysis, we observed that overall in Sweden, one-year mortality after hip-fracture at 25.1% was rather high during the period 2011–2014. We also observed that the average one-year adjusted mortality varied between 21.7% and 26.5% for hospitals and between 18.9% and 29.5% for municipalities. However, while the patient socioeconomic and clinical characteristics appears relevant for predicting mortality, only a minor part of the patient variation was explained by the hospital and municipality levels. Therefore, future interventions should focus on identifying high risk patient groups and be nationwide rather than directed at specific hospital or municipalities.

## Supporting information

**S1 Table. Sociodemographic risk score.**
(DOCX)

**S2 Table. Clinical risk score.**
(DOCX)

**S1 File. Statistical appendix.**
(DOCX)

## Acknowledgments

We thank the staff of the hospital departments caring for patients with hip fracture for their continuous effort and contribution to acquisition of the data in the registries.

## Author Contributions

**Conceptualization:** Pia Kjær Kristensen, George Leckie.

**Data curation:** Raquel Perez-Vicente, Juan Merlo.

**Formal analysis:** Raquel Perez-Vicente.

**Funding acquisition:** Pia Kjær Kristensen, Juan Merlo.

**Methodology:** Pia Kjær Kristensen, George Leckie, Juan Merlo.

**Project administration:** Pia Kjær Kristensen, Juan Merlo.

**Resources:** Juan Merlo.

**Software:** Raquel Perez-Vicente, George Leckie, Juan Merlo.

**Supervision:** Søren Paaske Johnsen, Juan Merlo.

**Validation:** Pia Kjær Kristensen, Raquel Perez-Vicente, George Leckie, Søren Paaske Johnsen, Juan Merlo.

**Visualization:** Pia Kjær Kristensen, Juan Merlo.

**Writing – original draft:** Pia Kjær Kristensen, Juan Merlo.

**Writing – review & editing:** Pia Kjær Kristensen, Raquel Perez-Vicente, George Leckie, Søren Paaske Johnsen, Juan Merlo.

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
