## [Decision Letter · Decision Letter 0]

7 Apr 2020

PONE-D-20-07054

Disentangling the contribution of hospitals and municipalities for understanding patient level differences in one-year mortality risk after hip-fracture: a cross-classified multilevel analysis in Sweden

PLOS ONE

Dear Dr. Kristensen,

Thank you for submitting your manuscript to PLOS ONE. After careful consideration, we feel that it has merit but does not fully meet PLOS ONE’s publication criteria as it currently stands. Therefore, we invite you to submit a revised version of the manuscript that addresses the points raised during the review process.

The study has been reviewed by one expert and the editor, who consider the study of merit and well presented. There are some specific suggestions to be considered by the authors, including the possibility of risk clustering and how this could affect the analysis. Further, additional descriptive parameters of the population could be of interest for the analysis.

We would appreciate receiving your revised manuscript by May 22 2020 11:59PM. To enhance the reproducibility of your results, we recommend that if applicable you deposit your laboratory protocols in protocols.io, where a protocol can be assigned its own identifier (DOI) such that it can be cited independently in the future. For instructions see: http://journals.plos.org/plosone/s/submission-guidelines#loc-laboratory-protocols

We look forward to receiving your revised manuscript.

Kind regards,

Pablo Garcia de Frutos

Academic Editor

PLOS ONE

Journal Requirements:

"we thank the Health Research Fund of Central Denmark Region Denmark and the Swedish Research Council for supporting this work https://www.swecris.se/betasearch/details/project/201701321VR and Vetenskapsrådet projekt-id: 2017-01321."

"Juan Merlo

Vetenskapsrådet projekt-id: 2017-01321

https://www.swecris.se/betasearch/details/project/201701321VR

NO

Pia Kjær Kristensen

The Health Research Fund of Central Denmark Region

A869

https://www.rm.dk/sundhed/faginfo/forskning/region-midtjyllands-sundhedsvidenskabelige-forskningsfond/

NO"

Reviewers' comments:

Reviewer's Responses to Questions

**Comments to the Author**

1. Is the manuscript technically sound, and do the data support the conclusions?

Reviewer #1: Yes

2. Has the statistical analysis been performed appropriately and rigorously? 

Reviewer #1: Yes

3. Have the authors made all data underlying the findings in their manuscript fully available?

Reviewer #1: No

4. Is the manuscript presented in an intelligible fashion and written in standard English?

Reviewer #1: Yes

5. Review Comments to the Author

Reviewer #1: This paper reports an analysis of national data on hip fracture mortality in older adults using a cross-classified multilevel framework to examine potential cluster effects by hospital and by municipality. Strengths of this study include the use of a large and comprehensive dataset and a sophisticated analytical approach. Overall, this is a strong contribution and my comments are relatively minor.

1. Additional supplementary information regarding the calculation of the sociodemographic and clinical risk scores would be valuable -- e.g., results for the logistic regression models from which these scores were derived would be good to include in the supplemental materials, including model fit and the parameter estimates used to compute the individual risk scores.

2. Because the cluster VPC statistics were calculated only after controlling for patient sociodemographic and clinical factors, there is some risk of under-estimating geographic disparities due to clustering of sociodemographic risk factors and patient comorbidity rates. It is likely that these risk factors are themselves clustered by municipality (and potentially to a lesser extent by hospital) based on local deprivation. The analytic reasons for taking the approach you used is explained clearly, but the issue of risk clustering should be briefly addressed in the discussion section. In particular, the suggestion that interventions aimed at improving mortality outcomes should focus on individual factors at a national level rather than on focusing on particular low-performing hospitals and municipalities should be somewhat tempered by the fact that certain specific places may require more focussed intervention because they include clusters of high-risk patients.

6. PLOS authors have the option to publish the peer review history of their article (what does this mean?). If published, this will include your full peer review and any attached files.

Reviewer #1: No

---

## [Author Response · Author response to Decision Letter 0]

30 Apr 2020

Reply: We thank for pointing out this example of author guidelines. On the title page we have added the ¶ symbol for 1st set of equal contributors. We have corrected the headings to be written in sentence case instead of capitalize. In addition, we have deleted the graphic images and space in the tables. 

2. We note that you have indicated that data from this study are available upon request. PLOS only allows data to be available upon request if there are legal or ethical restrictions on sharing data publicly. For information on unacceptable data access restrictions, please see http://journals.plos.org/plosone/s/data-availability#loc-unacceptable-data-access-restrictions. In your revised cover letter, please address the following prompts:

Reply: We apologize for the lacking information on data approval and access. The dataset are only available from the Swedish National Board of Health and Welfare and Statistics Sweden upon request following an ethical approval. The database was constructed after approval from the Ethical Committee in Sweden (https://etikprovningsmyndigheten.se/) and from the data safety committees of the Swedish National Board of Health and Welfare (https://www.socialstyrelsen.se/) and Statistics Sweden (https://www.scb.se/en/).

"we thank the Health Research Fund of Central Denmark Region Denmark and the Swedish Research Council for supporting this work https://www.swecris.se/betasearch/details/project/201701321VR and Vetenskapsrådet projekt-id: 2017-01321." We note that you have provided funding information that is not currently declared in your Funding Statement. However, funding information should not appear in the Acknowledgments section or other areas of your manuscript. We will only publish funding information present in the Funding Statement section of the online submission form. Please remove any funding-related text from the manuscript and let us know how you would like to update your Funding Statement. Currently, your Funding Statement reads as follows:

"Juan Merlo

Vetenskapsrådet projekt-id: 2017-01321

https://www.swecris.se/betasearch/details/project/201701321VR

NO

Pia Kjær Kristensen

The Health Research Fund of Central Denmark Region

A869

https://www.rm.dk/sundhed/faginfo/forskning/region-midtjyllands-sundhedsvidenskabelige-forskningsfond/

NO"

Reply: We thank the editor for pointing this out. We have deleted the phrases regarding funding statement in the acknowledgement section in the manuscript page 19 from line 340 to line 343. The original expression in the acknowledgement section (“the Swedish Research Council for supporting this work https://www.swecris.se/betasearch/details/project/201701321VR and Vetenskapsrådet projekt-id: 2017-01321”) was unfortunately confusing as it may represent two different funding statements. However, the “Swedish Research Council” is the English name of Swedish “Vetenskapsrådet”. So the funding statement for Juan Merlo is: 

Swedish Research Council (i.e., Vetenskapsrådet) project-id: 2017-01321. https://www.swecris.se/betasearch/details/project/201701321VR

Reviewer #1: This paper reports an analysis of national data on hip fracture mortality in older adults using a cross-classified multilevel framework to examine potential cluster effects by hospital and by municipality. Strengths of this study include the use of a large and comprehensive dataset and a sophisticated analytical approach. Overall, this is a strong contribution and my comments are relatively minor.

Reply: We thank the reviewer for the kind remarks.

1. Additional supplementary information regarding the calculation of the sociodemographic and clinical risk scores would be valuable -- e.g., results for the logistic regression models from which these scores were derived would be good to include in the supplemental materials, including model fit and the parameter estimates used to compute the individual risk scores.

Reply: We agree with the reviewer that calculation of the sociodemographic and clinical risk scores would be valuable for the interested reader. We have, therefore, as suggested by the reviewer, updated the Statistical appendix (labeled S1-S2 Table) to include information on the beta coefficients for the included variables in the Socioeconomic and clinic risk score equations as well as proving AUC values. 

2. Because the cluster VPC statistics were calculated only after controlling for patient sociodemographic and clinical factors, there is some risk of under-estimating geographic disparities due to clustering of sociodemographic risk factors and patient comorbidity rates. It is likely that these risk factors are themselves clustered by municipality (and potentially to a lesser extent by hospital) based on local deprivation. The analytic reasons for taking the approach you used is explained clearly, but the issue of risk clustering should be briefly addressed in the discussion section. In particular, the suggestion that interventions aimed at improving mortality outcomes should focus on individual factors at a national level rather than on focusing on particular low-performing hospitals and municipalities should be somewhat tempered by the fact that certain specific places may require more focussed intervention because they include clusters of high-risk patients.

Reply: We thank the reviewer for bringing our attention to the possible risk of under-estimating geographic disparities due to clustering of sociodemographic risk factors and patient comorbidity rates. We have now estimated the VPC statistics for the empty model for considering the maximal clustering. The empty model had a VPC values for both hospitals and municipalities of 0.007. We therefore, think that the risk of under-estimating geographic disparities is minor. Besides, even if the empty VPC were much higher and reduced when considering the composition of patients (because some hospitals include clusters of high-risk patients), the directing resources to high-risk patients on a national basis would automatically allocate more resources to those hospitals and municipalities with more high-risk patients. We have therefore, elaborated the paragraph on page 17 from line 312 to line 318 in the Discussion section to the flowing: 

We found that hospital and mortality differences in average mortality risks were only a minor share of the total individual variance in the propensity to die or that, in other words, adding the hospital and municipality level does not increase the discriminatory accuracy obtained by patient level information only. This means that the hospital and municipality performances are homogenous overall in the country. Therefore, our results indicate that we will not lower mortality after hip fracture in Sweden by focusing on specific hospitals or specific municipalities with high average mortality. Rather, special efforts to reduce mortality after hip fracture should be focused on vulnerable patient groups of hip fracture patients wherever they are. From this reason, however, hospitals and municipalities with a higher number of vulnerable patients than other will need more intense interventions.

---

## [Decision Letter · Decision Letter 1]

19 May 2020

Disentangling the contribution of hospitals and municipalities for understanding patient level differences in one-year mortality risk after hip-fracture: a cross-classified multilevel analysis in Sweden

PONE-D-20-07054R1

Dear Dr. Kristensen,

We are pleased to inform you that your manuscript has been judged scientifically suitable for publication and will be formally accepted for publication once it complies with all outstanding technical requirements.

With kind regards,

Pablo Garcia de Frutos

Academic Editor

PLOS ONE

Additional Editor Comments (optional):

Reviewers' comments:

Reviewer's Responses to Questions

**Comments to the Author**

1. If the authors have adequately addressed your comments raised in a previous round of review and you feel that this manuscript is now acceptable for publication, you may indicate that here to bypass the “Comments to the Author” section, enter your conflict of interest statement in the “Confidential to Editor” section, and submit your "Accept" recommendation.

Reviewer #1: All comments have been addressed

2. Is the manuscript technically sound, and do the data support the conclusions?

Reviewer #1: Yes

3. Has the statistical analysis been performed appropriately and rigorously? 

Reviewer #1: Yes

4. Have the authors made all data underlying the findings in their manuscript fully available?

Reviewer #1: Yes

5. Is the manuscript presented in an intelligible fashion and written in standard English?

Reviewer #1: Yes

6. Review Comments to the Author

Reviewer #1: (No Response)

7. PLOS authors have the option to publish the peer review history of their article (what does this mean?). If published, this will include your full peer review and any attached files.

Reviewer #1: No

---

## [Editor Report · Acceptance letter]

22 May 2020

PONE-D-20-07054R1 

Disentangling the contribution of hospitals and municipalities for understanding patient level differences in one-year mortality risk after hip-fracture: a cross-classified multilevel analysis in Sweden 

Dear Dr. Kristensen:

I am pleased to inform you that your manuscript has been deemed suitable for publication in PLOS ONE. Congratulations! Your manuscript is now with our production department. 

With kind regards,

on behalf of

Dr. Pablo Garcia de Frutos 

Academic Editor

PLOS ONE